# Factors Influencing Nursing Students’ Academic Engagement and Burnout During the COVID-19 Pandemic: A Path Analysis

**DOI:** 10.3390/nursrep15090339

**Published:** 2025-09-18

**Authors:** Ji Hyun Park, Jin-Hwa Park

**Affiliations:** College of Nursing, Daegu Catholic University, Daegu 42472, Republic of Korea; jhpark0917@cu.ac.kr

**Keywords:** academic burnout, academic engagement, nursing, stress, students, COVID-19

## Abstract

**Background/Objectives**: Increases in unemployment due to the COVID-19 pandemic and doctors’ strike have intensified job-seeking stress among nursing students, contributing to academic pressure, increased stress levels, reduced participation, and a greater risk of burnout. This study investigates the relationships between job-seeking stress, academic self-efficacy, professor trust, academic engagement, and academic burnout through path analysis. **Methods**: A total of 496 nursing students enrolled in four-year nursing programs in South Korea participated in this study. Data were gathered using structured questionnaires from 30 August to 13 December 2021. **Results**: Job-seeking stress, academic self-efficacy, and professor trust significantly influenced academic engagement, accounting for 37.2% of its variance. Academic burnout was primarily explained by job-seeking stress and academic self-efficacy, with an explanatory power of 50.4%. Furthermore, academic self-efficacy played a mediating role in the relationships between job-seeking stress and both academic engagement and burnout. **Conclusions**: Developing interventions to enhance academic self-efficacy is crucial. Additionally, strategies should be implemented to alleviate job-seeking stress, foster academic engagement, and reduce the risk of burnout among nursing students.

## 1. Introduction

College students in South Korea have various sources of stress—academic, interpersonal, extracurricular, family-related, and job-related—but it is known that stress related to job preparation is the most stressful [1]. The hiring of new university graduates has sharply declined due to the impact of the COVID-19 pandemic [2]. The recruitment of new registered nurses is no exception [3]. Job-seeking stress has been found to be associated with a variety of health-related problems, including tinnitus, nervousness, depression, gastrointestinal disturbances, insomnia, and social phobia, with approximately 70% of adults reportedly experiencing one or more of these symptoms [4]. This also affects students’ academic engagement and contributes to academic burnout. To secure positions at highly competitive institutions, such as tertiary hospitals, Korean nursing students must dedicate substantial time and effort to achieving excellence in both theoretical coursework and clinical practicums. Along with the burden of having to pass the national nursing licensure exam, they are expected to cultivate proficiency in professional English and actively engage in volunteer activities and international training programs to enhance their professional competitiveness in an increasingly demanding job market [5]. Moreover, in response to the South Korean government’s proposal to increase medical school admissions to address a doctor shortage, over 12,000 junior doctors walked out, medical professors joined in solidarity, and more than 95% of 2025 medical graduates are reportedly refusing to take the licensing exam [6]. In the first half of 2024, university hospitals faced severe financial difficulties due to the medical resident strike, resulting in almost no recruitment of new registered nurses. Despite an estimated 21,000 fourth-year nursing students nationwide, only one university hospital among the tertiary hospitals in the Seoul metropolitan area recruited new registered nurses during this period [7]. The doctors’ strike has exacerbated job-seeking anxiety and stress among nursing students. An appropriate stress level positively affects academic performance. However, excessive stress causes academic burnout, including constant fatigue, emotional exhaustion, and a cynical attitude toward the academic process [8].

Furthermore, nursing students’ academic burnout can affect clinical nursing performance after graduation [9]. It affects major satisfaction [10]; decreased academic achievement affects nursing students’ ability to perform nursing tasks and lowers their occupational preparedness after graduation [11]. They may experience complications such as sleep disorders, including insufficient sleep, resulting in physical and psychological problems [12]. During the COVID-19 pandemic, non-face-to-face teaching methods were introduced to prevent the spread of infection in educational institutions. Remote teaching methods were difficult for instructors and learners to adapt to, causing increased burnout among nursing students [13]. Academic burnout leads to poor academic participation and engagement in school life [14].

Academic engagement is the degree to which learners actively participate in learning activities and processes to achieve desired learning outcomes [15]. It is related to positive outcomes such as academic achievement and continuity. Academic engagement increases with sufficient teacher support [16]. Moreover, academic self-efficacy refers to an individual’s confidence in their ability to accomplish specific tasks at the expected level [17]. It positively influences academic achievement and major satisfaction [18,19] in nursing students. Although academic burnout and engagement are theoretically regarded as opposite concepts [8], they occur simultaneously in academia. Thus, it is necessary to examine them together.

This study selected job-seeking stress, academic self-efficacy, and professor trust as key variables that affect nursing students’ academic burnout and engagement. Therefore, this study aimed to examine the effects of job stress on academic burnout and engagement in nursing students. We investigated the antecedent variables that affect their academic participation and burnout by constructing and verifying a path model. This could provide basic data for various educational interventions to reduce nursing students’ academic burnout and increase their academic participation by suggesting causal relationships among variables.

## 2. Materials and Methods

### 2.1. Design

This study used path analysis to explore the relationship between job-seeking stress, academic self-efficacy, professor trust, academic burnout, and academic engagement in nursing students. We derived a hypothetical model for the causal relationship between these variables and tested the proposed hypotheses.

### 2.2. Hypotheses

Group 1: Job-seeking stress
1.Nursing students’ job-seeking stress will negatively affect academic engagement.2.Nursing students’ job-seeking stress will positively affect academic burnout.3.Nursing students’ job-seeking stress will negatively affect academic self-efficacy.4.Nursing students’ job-seeking stress will negatively affect professor trust.Group 2: Academic self-efficacy
5.Nursing students’ academic self-efficacy will positively affect academic engagement.6.Nursing students’ academic self-efficacy will negatively affect academic burnout.Group 3: Trust in professors
7.Nursing students’ trust in professors will positively affect academic self-efficacy.8.Nursing students’ trust in professors will positively affect academic engagement.9.Nursing students’ trust in professors will negatively affect academic burnout.

### 2.3. Participants

The participants were nursing students enrolled in 4-year nursing programs across 11 nursing departments nationwide. The minimum sample size required for correlation analysis with a significance level of 0.05, power of 0.90, and a medium effect size (0.20) was calculated to be 258 using G*Power 3.1.9.2. A total of 450 questionnaires were distributed, and 417 were returned (92.7%). After excluding four questionnaires with incomplete responses, 413 were retained for the final analysis, yielding a valid response rate of 91.8%.

### 2.4. Ethical Considerations

For the ethical protection of the research participants, this research was approved by the institutional review board of the researcher’s affiliated institution (CUIRB-2021-0038). Before starting the survey, participants were informed about the research purpose and that the research results will not be used for any purpose other than the research itself. The anonymity guarantee for research participation and possible benefits and disadvantages were explained. It was also explained that all research materials are encrypted in each document, stored in a device accessible only to the research director for three years, and then safely discarded afterward. Mobile gift certificates were provided to study participants as a token of appreciation.

### 2.5. Measurements

#### 2.5.1. General Characteristics

The general characteristics in this study were sex, age, grade, regular exercise, perceived stress, perceived interpersonal relationships, nursing major satisfaction, clinical practicum satisfaction, career plan after graduation, job-seeking concerns, job-seeking stress, and changes in study concentration after the COVID-19 pandemic.

#### 2.5.2. Job-Seeking Stress

Job stress was measured using the Korean Career Stress Inventory developed by Choi et al. [20] to measure job stress for Korean college students. This instrument comprises four sub-factors with five questions each about career ambiguity, lack of information, job pressure, and external conflict. Each item is ranked on a 5-point Likert scale, ranging from 1 (“very low stress”) to 5 (“very high stress”), with higher scores indicating higher job stress levels. In a study by Choi et al. [20], overall Cronbach’s ⍺ was 0.89, and sub-factors had the values ⍺ = 0.89, ⍺ = 0.84, and ⍺ = 0.82 for lack of information, job pressure, and external conflict, respectively. In this study, overall Cronbach’s ⍺ was 0.92, and sub-factors had the values ⍺ = 0.89, ⍺ = 0.88, ⍺ = 0.85, and ⍺ = 0.84 for career ambiguity, lack of information, job pressure, and external conflict, respectively.

#### 2.5.3. Academic Self-Efficacy

Academic self-efficacy was measured using the Academic Self-efficacy Scale developed and validated by Kim and Park [21]. This instrument comprises 28 items with 3 sub-factors: self-confidence (8 items), self-regulation efficacy (10 items), and task difficulty preference (10 items). Each item is scored on a 6-point Likert scale, ranging from 1 (“not at all”) to 6 (“strongly agree”), with higher scores indicating higher academic self-efficacy. In a study [21], the sub-factors’ reliability values were ⍺ = 0.74, ⍺ = 0.76, and ⍺ = 0.84 for self-confidence, self-regulation efficacy, and preference for task difficulty, respectively. In this study, overall Cronbach’s ⍺ was 0.89, and sub-factors had the values ⍺ = 0.90, ⍺ = 0.90, and ⍺ = 0.85 for self-confidence, self-regulation efficacy, and task difficulty preference, respectively.

#### 2.5.4. Professor Trust

Professor trust was measured using the Professor Trust Scale developed by Jeong and Park [22]. This instrument comprises 27 questions with 4 sub-factors: intimacy (8 questions), professionalism (8 questions), lecture ability (5 questions), and leadership (6 questions). Each question is scored on a 5-point Likert scale, ranging from 1 (“not at all”) to 5 (“strongly agree”). In the original study [22], overall Cronbach’s ⍺ was 0.96, and sub-factors had the values ⍺ = 0.85, ⍺ = 0.83, ⍺ = 0.84, and ⍺ = 0.80 for intimacy, expertise, teaching, and leadership, respectively. In this study, overall Cronbach’s ⍺ was 0.97, and sub-factors had the values ⍺ = 0.88, ⍺ = 0.95, ⍺ = 0.92, and ⍺ = 0.92 for intimacy, expertise, teaching, and leadership, respectively.

#### 2.5.5. Academic Engagement

Academic engagement was measured using the 13-item Utrecht Work Engagement Scale—Student (UWES-S) tool validated by Choo and Sohn [23] for Korean university students. The UWES-S was originally a 17-item instrument developed by Schaufeli et al. [8]. It comprises three sub-factors: vigor (6 items), dedication (3 items), and absorption (4 items). Each item is scored on a 6-point Likert scale, ranging from 1 (“not at all”) to 7 (“always”), where the higher the score, the higher the degree of academic engagement. In the original study [8], Cronbach’s ⍺ was 0.63–0.81 for each sub-factor; additionally, it was 0.77–0.82 in the study by Choo and Sohn [23]. In this study, overall Cronbach’s ⍺ was 0.92, and sub-factors had the values ⍺ = 0.87, ⍺ = 0.81, and ⍺ = 0.89 for vigor, dedication, and absorption, respectively.

#### 2.5.6. Academic Burnout

Academic burnout was measured using the Maslach Burnout Inventory—Student Survey (MBI-SS) validated by Lee and Lee [24] for Korean medical students. It was developed by Schaufeli et al. [8] to measure students’ burnout levels. This instrument comprises 14 items with 3 sub-factors: exhaustion (5 items), inefficacy (5 items), and cynicism (4 items). Each item is scored on a 6-point Likert scale, ranging from 1 (“not at all”) to 6 (“always”), with higher scores indicating more severe academic burnout. In the study of Lee and Lee [24], Cronbach’s ⍺ values were 0.84, 0.77, and 0.84 for exhaustion, inefficacy, and cynicism, respectively. In this study, overall Cronbach’s ⍺ was 0.83, and sub-factors had the values ⍺ = 0.84, ⍺ = 0.83, and ⍺ = 0.79 for exhaustion, inefficacy, and cynicism, respectively.

### 2.6. Data Collection

Data were collected between 30 August and 13 December 2021, using convenience sampling in the 11 nursing departments. After obtaining cooperation from faculty in each department, a survey package—including the questionnaire, instructions for completion, and a return envelope—was either mailed or handed directly to participants. Students who consented to participate were asked to complete the survey, seal it in the envelope, and return it to the research team. Completion of the questionnaire required approximately 20 min. Returned surveys were screened for completeness, and only fully completed questionnaires were included in the analysis.

### 2.7. Data Analysis

The collected data were analyzed using IBM SPSS Statistics Version 25 and AMOS Version 20 (IBM Corp., Armonk, NY, USA). Descriptive statistics were used to summarize the participants’ general characteristics, as well as the distribution of key study variables. To examine differences in academic engagement and burnout across demographic subgroups, independent t-tests or one-way ANOVAs were applied, with Scheffé’s post hoc test conducted when group differences were significant. These procedures were chosen to identify potential variations in outcomes according to categorical characteristics. Finally, path analysis was conducted to test the hypothesized model, as it allows for simultaneous estimation of direct and indirect effects among multiple variables.

## 3. Results

### 3.1. General Characteristics

A total of 396 participants were enrolled in this study: 49 males (12.4%) and 347 females (87.6%). Participants’ average age was 22.24 ± 1.38 years. Stress levels were moderate for 233 participants (58.8%), high for 131 (33.1%), and low for 32 (8.1%). Nursing major satisfaction was 6.63 ± 1.93 and clinical practicum satisfaction averaged 6.44 ± 1.97 out of 10 points. Regarding career plans after graduation, 320 (80.8%) participants planned to work in university hospitals. Furthermore, 312 (78.8%) said that COVID-19 affected their ability to concentrate on their studies (Table 1).

### 3.2. Scores for Job-Seeking Stress, Academic Self-Efficacy, Professor Trust, Academic Engagement, and Academic Burnout

The mean score for job-seeking stress was 2.78 ± 0.67 from 1 to 5 points, academic self-efficacy was 3.70 ± 0.59 from 1 to 6 points, professor trust was 4.00 ± 0.59 from 1 to 5 points, academic engagement was 2.83 ± 0.68 from 1 to 5 points, and academic burnout was 3.72 ± 0.64 from 1 to 7 points (Table 2).

### 3.3. Differences Between Academic Engagement and Burnout According to General Characteristics

Academic engagement according to participants’ general characteristics was statistically significant in relation to regular exercise (t = 2.26, *p* = 0.025), perceived stress levels (F = 6.31, *p* = 0.002), perceived interpersonal relationships (F = 4.76, *p* = 0.003), and post-graduation career plans (F = 4.29, *p* = 0.002). Participants who exercised regularly showed significantly higher academic engagement. However, post hoc analysis revealed that differences in perceived stress levels were not statistically significant. Academic engagement was also significantly higher among participants who rated their interpersonal relationships as “very good” or “good”. Regarding career plans after graduation, participants intending to work at a university hospital had significantly higher academic engagement compared to those planning to attend graduate school.

Academic burnout, according to the participants’ general characteristics, was statistically significant in terms of sex (t = 2.71, *p* = 0.007), age (F = 6.66, *p* = 0.001), whether they exercised regularly (t = 3.10, *p* = 0.002), degree of perceived stress (F = 13.86, *p* < 0.001), degree of perceived interpersonal relationships (F = 6.83, *p* < 0.001), and career plans after graduation (F = 3.81, *p* = 0.005). Academic burnout was significantly higher among female participants and those under 23 years of age compared to participants over 23. In the case of not exercising regularly, academic burnout was statistically significantly higher. Post hoc analysis revealed that those who reported high stress levels had significantly higher academic burnout than those with average or low stress levels. Those who answered that the degree of perceived interpersonal relationship was average or poor had statistically significantly higher academic burnout than those who answered that it was good or very good. The post hoc analysis results of career plans after graduation were not statistically significant (Table 1).

### 3.4. Testing the Path Model

To identify the mediating effect of academic self-efficacy and professor trust on academic engagement and academic burnout, job-seeking stress was used as an exogenous variable; academic self-efficacy and professor trust were used as endogenous and exogenous variables, respectively; and academic engagement and academic burnout were used as endogenous variables. The normality of the data was reviewed to apply the maximum likelihood method to the hypothetical model test. For each variable, the absolute value of skewness was 0.08–1.30, which was 3 or less, and the absolute value of kurtosis was 0.001–4.67, which was 10 or less. The maximum likelihood method could be used to test the model’s fitness [25] (Table 2).

### 3.5. Testing and Modification of Hypothetical Models

To assess how well the proposed model fits the observed data, we examined a range of model fit indices. For absolute fit, we considered the normed chi-square (χ^2^/df), which evaluates model fit while accounting for model complexity—values below 3 are typically considered acceptable. We also included the Goodness of Fit Index (GFI), which indicates how closely the model reproduces the observed covariance matrix (values closer to 1 suggest better fit), and the Standardized Root Mean Square Residual (SRMR), which measures the average discrepancy between observed and predicted correlations—lower values indicate a better fit. Additionally, we used the Root Mean Square Error of Approximation (RMSEA) to assess how well the model would fit the population covariance matrix; values below 0.05 indicate close fit, and values up to 0.08 suggest reasonable fit. To evaluate incremental fit, we employed the Tucker–Lewis Index (TLI), Normed Fit Index (NFI), and Comparative Fit Index (CFI), which compare the target model with a baseline model, typically the null model. Values above 0.90 or 0.95 are generally considered to reflect a good fit. Lastly, for parsimonious fit, which considers both the goodness of fit and model simplicity, we examined the Adjusted Goodness of Fit Index (AGFI), a modified version of GFI that penalizes model complexity. Higher AGFI values indicate a better balance between fit and simplicity [25].

Initially, the results indicated poor model fit, with normed χ^2^ = 51.80 (*p* < 0.001), GFI = 0.95, SRMR = 0.050, RMSEA = 0.359, TLI = 0.197, NFI = 0.919, CFI = 0.920, and AGFI = 0.297. The notably high normed χ^2^ and RMSEA values suggested substantial discrepancies between the hypothesized model and the data, requiring further adjustments. Given these poor fit indicators, we revised the model based on modification indices. Specifically, we deleted two insignificant paths: professor trust leading to academic burnout, and professor trust leading to academic participation. Additionally, correlations between the measurement errors of academic engagement and academic burnout, both endogenous latent variables, were established based on the modification indices.

After these adjustments, a difference test in model fit was conducted to compare the original hypothetical model and the modified model. The modified model showed a substantially improved and acceptable fit with normed χ^2^ = 1.71 (*p* = 0.182), GFI = 0.997, SRMR = 0.020, RMSEA = 0.042, TLI = 0.989, NFI = 0.995, CFI = 0.998, and AGFI = 0.974. These revised results indicated that the final modified model satisfactorily met recommended criteria for good fit (Table 3).

### 3.6. Testing Modified Models

After testing the hypotheses established in this research model using the path coefficients between the variables, the following hypotheses were adopted: Hypothesis 1. Nursing students’ job-seeking stress will negatively affect academic engagement (Estimate = −0.116, CR = −2.589, *p* = 0.010). Hypothesis 2. Nursing students’ job-seeking stress will positively affect academic burnout (Estimate = 0.315, CR = 7.901, *p* < 0.001). Hypothesis 3. Nursing students’ job-seeking stress will negatively affect academic self-efficacy (Estimate = −0.428, CR = −9.690, *p* < 0.001). Hypothesis 4. Nursing students’ job-seeking stress will negatively affect professor trust (Estimate = −0.169, CR = −3.399, *p* < 0.001). Hypothesis 5. Nursing students’ academic self-efficacy will positively affect academic engagement (Estimate = 0.547, CR = 12.168, *p* < 0.001). Hypothesis 6. Nursing students’ academic self-efficacy will negatively affect academic burnout (Estimate = −507, CR = −12.700, *p* < 0.001). Hypothesis 7. Nursing students’ trust in professors will positively affect academic self-efficacy (Estimate = 0.198, CR = 4.482, *p* < 0.001). Hypothesis 8 and 9 were rejected.

As a result of testing the path to identifying the relationship between variables, job-seeking stress had a significant direct effect on academic self-efficacy (β = −0.428, *p* < 0.001), professor trust (β = −0.169, *p* = 0.001), academic engagement (β = −0.116, *p* = 0.023), and academic burnout (β = 0.315, *p* < 0.001). Academic self-efficacy had a significant direct effect on academic burnout (β = −507, *p* < 0.001). The direct (β = −0.116, *p* = 0.023) and indirect (β = −0.252, *p* < 0.001) effects of job-seeking stress on academic engagement were significant. Academic self-efficacy had a significant direct effect (β = 0.547, *p* < 0.001) on academic engagement, indicating that academic self-efficacy had a partial mediating effect between job-seeking stress and academic engagement. Moreover, job-seeking stress had significant effects on both the direct (β = −0.428, *p* < 0.001) and indirect (β = −0.033, *p* < 0.001) effects of academic self-efficacy. Professor trust had a significant direct effect (β = 0.198, *p* < 0.001) on academic self-efficacy; in the effect of job-seeking stress on academic self-efficacy, professor trust showed a partial mediating effect.

In this study model, the squared multiple correlations (SMC) of academic self-efficacy, professor trust, academic engagement, and academic burnout were 0.251 (25.1% of the explanatory power), 0.028 (2.8% of the explanatory power), 0.372 (37.2% of the explanatory power), and 0.504 (50.4% of explanatory power), respectively (Table 4, Figure 1).

## 4. Discussion

This study aimed to identify the factors affecting nursing students’ academic engagement and burnout by testing the paths between variables and increase academic engagement and lower academic burnout. In the modified path model, 37.2% of academic engagement was explained by job-seeking stress, academic self-efficacy, and professor trust. Moreover, 50.4% of academic burnout was explained by job-seeking stress, academic self-efficacy, and professor trust.

In this study, the mean academic engagement of nursing students was 2.83 out of 5. This was lower than the average of 3.07 points of academic engagement among Korean nursing students measured with the same instrument [15]. The learning participation of Korean nursing students measured with other tools was an average of 3.28 out of 5 points [26], similar to the score of 3.83 out of 7 for overseas nursing students [27]. This was assumed to be related to the COVID-19 pandemic because students had to adapt to online learning. This was understandably different from the previous face-to-face class methods. This contributed to students being more distracted during class; additionally, they experienced stress, anxiety, and helplessness [27].

The findings showed that nursing students’ academic burnout averaged 3.72 out of 7 points and 52.10 out of 98 total. In previous studies, academic burnout in nursing students was 33.15 out of 60 [13], 3.99 out of 5 [14], 70.26 out of 125 [9], and 2.97 out of 5 [27]. Thus, our findings were moderate or higher than those of the studies mentioned. A longitudinal prospective study conducted among Swedish nursing students found that academic burnout predicted lower learning engagement and job readiness during their studies, as well as reduced occupational skills and increased turnover intention one year after graduation [11]. Based on this, it can also be inferred that nursing students’ academic burnout affects their current academic engagement and their career identity after becoming a nurse. Therefore, developing a plan to address academic burnout in nursing students is imperative. Given the pressure of passing the licensure exam and obtaining a job simultaneously, specific learning strategies should be developed to maximize students’ academic engagement and alleviate their academic burnout.

Nursing students’ job-seeking stress in this study was 2.78 out of 5. According to previous studies, nursing students’ job-seeking stress was lower than ours, for example, 2.10 out of 5 [28] and 2.47 out of 5 [29]. Following the World Health Organization’s declaration of COVID-19 as a global pandemic in March 2020, the labor market experienced a significant downturn, including a decline in nursing employment opportunities, which in turn intensified job-seeking stress among nursing students. In a previous study, nursing students also showed anxiety and stress about whether they would be able to secure a job without adequate clinical practicum [3]. As the pandemic continues, uncertainty about the future grows, and job seekers are experiencing considerably high anxiety levels about decreasing jobs, exacerbating job-seeking stress. Thus, the rate of experiencing mental symptoms such as depression, anxiety, and anger is increasing [30]. The job-seeking stress of college students has also increased compared to that before the COVID-19 pandemic [3].

In this study, nursing students’ academic self-efficacy was 3.70 out of 6 points. This was similar to the results obtained by measuring nursing students’ academic self-efficacy using the same tool, with a score of 3.70 [14]. Our study revealed that academic self-efficacy among nursing students had a direct impact on both academic engagement and academic burnout. This was consistent with the results of previous research on academic self-efficacy showing a positive relationship with learning engagement [26]. Therefore, increasing nursing students’ academic self-efficacy can increase academic engagement and reduce academic burnout. In this study, academic self-efficacy was directly affected by job-seeking stress, and professor trust showed a partial mediating effect in the effect of job-seeking stress on academic self-efficacy. It is difficult to make a direct comparison because, to the best of our knowledge, no studies have explored the relationship between nursing students’ job-seeking stress and professor trust. However, a previous study suggested that nursing students’ professor trust mediates the relationship between academic stress and college life adjustment; even if students experience job-seeking stress, professor trust could affect students’ academic self-efficacy.

Professor trust is a concept that includes intimacy, lecture ability, leadership, and the professor’s expertise in their field of study [22]. Several strategies can strengthen students’ trust in their professors, including increasing student–faculty interaction and devoting more time to lesson preparation to improve instructional quality. However, Hypotheses 8 and 9, which proposed that nursing students’ trust in professors would directly influence academic engagement and burnout, were not supported. A possible explanation is that professor trust does not directly affect engagement or burnout but instead operates indirectly through mediating factors such as academic self-efficacy. Indeed, our results showed that professor trust significantly predicted academic self-efficacy (β = 0.198, *p* < 0.001), which in turn strongly predicted engagement (β = 0.547, *p* < 0.001) and burnout (β = −0.507, *p* < 0.001). These findings suggest that professor trust may be better understood as fostering students’ confidence in their academic abilities, which subsequently shapes their engagement and susceptibility to burnout. This indirect relationship is consistent with prior research highlighting the mediating role of self-efficacy in educational contexts [31]. Accordingly, professor trust may be more accurately regarded as an antecedent to self-efficacy beliefs rather than a direct determinant of student outcomes. Further research is needed to examine additional psychological mediators or contextual moderators that may clarify these indirect effects.

This study has several methodological limitations that should be acknowledged. One limitation is that the use of convenience sampling from 11 nursing departments may limit the representativeness of the sample. Although participants were recruited from multiple institutions nationwide, the findings cannot be generalized to all nursing students in Korea. Another is that because data collection relied on self-administered questionnaires distributed with the cooperation of faculty members, there may have been selection bias, as students who were more motivated or interested in the study topic may have been more likely to participate. Finally, the self-report nature of the survey may have introduced response bias or social desirability bias, which could affect the accuracy of the reported data. These limitations should be considered when interpreting the findings and assessing their generalizability.

This study revealed several meaningful insights. The use of path analysis was particularly appropriate for this study, as it enabled the simultaneous examination of direct and indirect effects among job-seeking stress, academic self-efficacy, professor trust, academic engagement, and academic burnout. This approach provided a more comprehensive understanding of the interrelationships among the study variables than would have been possible with simpler statistical techniques. While job-seeking stress had a significant impact on both academic engagement and burnout, the role of professor trust was found to be indirect, mediated through academic self-efficacy. This contrasts with prior studies that found a more direct link between faculty support and student outcomes, suggesting that in the context of heightened job uncertainty, students may rely more on internal beliefs (self-efficacy) than external sources (professors). The findings align with previous research suggesting that self-efficacy is a strong predictor of academic performance, learning engagement, and stress resilience. However, the higher-than-expected burnout scores and lower engagement scores, especially when compared to pre-pandemic studies, point to a broader systemic disruption in nursing education during COVID-19. Non-face-to-face classes, reduced clinical exposure, and a bleak employment landscape all appear to contribute. Therefore, practical interventions should focus not only on cognitive and emotional support systems but also on reinforcing confidence-building through structured academic feedback and skill-based training. In addition, it is recommended that future studies use longitudinal designs to examine changes in burnout and engagement trends post-pandemic.

This study revealed that job-seeking stress affects academic engagement and burnout through academic self-efficacy and professor trust. It is necessary to encourage nursing students to have hope through their academic self-efficacy and professor trust. It can be a strategy to increase academic engagement and reduce academic burnout if the instructor supports nursing students to set clear and specific goals and provides feedback to enhance academic self-efficacy. Therefore, the study results can be used as evidence to develop educational and resource programs that increase academic self-efficacy and professor trust to increase nursing students’ academic engagement and reduce academic burnout for those experiencing job-seeking stress.

## 5. Conclusions

This study identified key factors influencing academic engagement and burnout among nursing students through path analysis. The proposed model demonstrated that job-seeking stress, academic self-efficacy, and trust in professors were significantly associated with students’ levels of academic engagement and burnout. As academic burnout continues to increase, strategies to enhance engagement and reduce burnout are essential for supporting students’ successful adjustment to academic life and future clinical environments.

Based on these findings, it is necessary to develop and implement intervention programs that strengthen nursing students’ academic self-efficacy. Such programs could include elements like peer mentoring and stress management education. Enhancing academic self-efficacy may contribute to increased engagement and reduced burnout over time. Furthermore, the results highlight the importance of the professor–student relationship. It is recommended that nursing educators reflect on and refine their instructional strategies to build trust and foster a supportive learning environment. Ultimately, such efforts may enhance students’ academic self-efficacy, thereby increasing academic engagement and reducing burnout. Evaluating the effectiveness of these approaches will be essential in promoting sustainable academic performance and long-term professional development.

## Figures and Tables

**Figure 1 nursrep-15-00339-f001:**
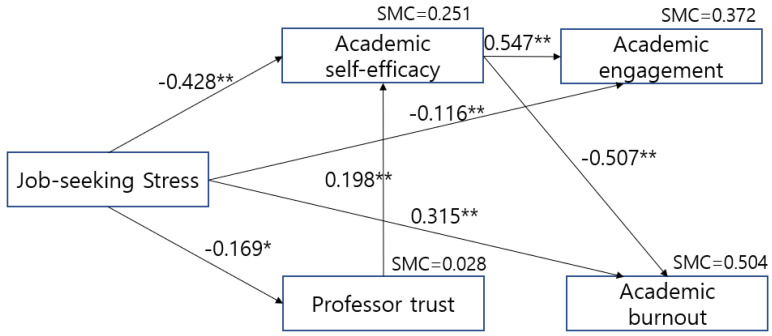
Revised model. *: *p* < 0.05, **: *p* < 0.001.

**Table 1 nursrep-15-00339-t001:** General characteristics of participants (*N* = 396).

Characteristics	Categories	n(%) or M ± SD	Academic Engagement		Academic Burnout	
			M ± SD	t/F(*p*) Scheffé	M ± SD	t/F(*p*) Scheffé
Sex	Male	49 (12.4)	2.85 ± 0.77	0.25 (0.804)	3.49 ± 0.62	2.71 (0.007)
	Female	347 (87.6)	2.82 ± 0.66		3.75 ± 0.63	
Age		22.24 ± 1.38				
	<22 ^a^	142 (35.8)	2.83 ± 0.72	1.66 (0.192)	3.82 ± 0.66	6.66 (0.001)
	22–<23 ^b^	127 (32.1)	2.74 ± 0.61		3.78 ± 0.63	a, b > c
	≥23 ^c^	127 (32.1)	2.90 ± 0.69		3.56 ± 0.59	
Grade	Sophomore	136 (34.3)	2.92 ± 0.75	0.06 (0.941)	3.78 ± 0.64	2.84 (0.061)
	Junior	133 (33.6)	2.88 ± 0.61		3.66 ± 0.65	
	Senior	127 (32.1)	2.90 ± 0.75		3.52 ± 0.60	
Regular exercise	Yes	129 (32.6)	3.06 ± 0.70	2.26 (0.025)	3.46 ± 0.68	3.10 (0.002)
	No	267 (67.4)	2.82 ± 0.70		3.75 ± 0.60	
Perceived stress level	Low ^a^	32 (8.1)	2.82 ± 0.95	6.31 (0.002)	3.23 ± 0.62	13.86 (<0.001)
	Moderate ^b^	233 (58.8)	3.04 ± 0.59		3.55 ± 0.59	a, b < c
	High ^c^	131 (33.1)	2.67 ± 0.78		3.95 ± 0.61	
Perceived interpersonal relationships	Very good ^a^	23 (5.8)	2.91 ± 0.98	4.76 (0.003)	3.45 ± 0.85	6.83 (<0.001)
	Good ^b^	252 (63.7)	2.90 ± 0.67	a, b > c, d	3.66 ± 0.63	a, b < c, d
	Average ^c^	113 (28.5)	2.68 ± 0.60		3.88 ± 0.55	
	Bad ^d^	8 (2.0)	2.27 ± 0.49		4.25 ± 0.38	
	Very bad	0 (0)	-		-	
Nursing major satisfaction		6.63 ± 1.93				
Clinical practicum satisfaction		6.44 ± 1.97				
Career plan after graduation	University hospitals ^a^	320 (80.8)	2.88 ± 0.66	4.29 (0.002)	3.66 ± 0.61	3.81 (0.005)
	General hospitals ^b^	40 (10.1)	2.68 ± 0.60	(a > e)	4.01 ± 0.69	
	Geriatric hospitals ^c^	7 (1.8)	2.24 ± 0.54		3.97 ± 0.50	
	Governments ^d^	25 (6.3)	2.69 ± 0.89		3.89 ± 0.79	
	Graduate school ^e^	4 (1.0)	1.94 ± 0.59		4.09 ± 0.48	
Shared concerns about finding a job	Friends	201 (50.8)	2.80 ± 0.67	1.59 (0.192)	3.72 ± 0.61	1.77 (0.153)
	Family	122 (30.8)	2.92 ± 0.66		3.66 ± 0.66	
	Alone	59 (14.9)	2.70 ± 0.76		3.88 ± 0.65	
	Others	14 (3.5)	2.86 ± 0.46		3.59 ± 0.69	
COVID-19 on job-seeking stress	Impacted	183 (46.2)	2.86 ± 0.69	0.75 (0.454)	3.70 ± 0.64	0.87 (0.384)
	Not-impacted	213 (53.8)	2.93 ± 0.72		3.62 ± 0.64	
COVID-19 on study concentration	Impacted	312 (78.8)	2.89 ± 0.67	0.43 (0.665)	3.68 ± 0.61	1.05 (0.296)
	Not-impacted	84 (21.2)	2.94 ± 0.82		3.57 ± 0.74	

The superscript numbers indicate the results of Scheffé’s post hoc test.

**Table 2 nursrep-15-00339-t002:** Degrees of job-seeking stress, academic self-efficacy, professor trust, academic engagement, and academic burnout (*N* = 396).

Variables	No. of Items (Categories)	Scale	M ± SD	Range	M/Item ± SD	Range (Item)	Skewness	Kurtosis
Job-seeking stress	20 (4)	5-point Likert	55.67 ± 13.38	20∼100	2.78 ± 0.67	1.00∼5.00	−0.08	−0.16
Academic self-efficacy	28 (3)	6-point Likert	103.28 ± 16.81	44∼155	3.70 ± 0.59	1.63∼5.54	0.20	0.31
Professor trust	27 (4)	5-point Likert	107.96 ± 15.92	29∼157	4.00 ± 0.59	1.07∼5.81	−1.30	4.67
Academic engagement	13 (3)	6-point Likert	36.73 ± 8.80	13∼59	2.83 ± 0.68	1.00∼4.54	−0.25	−0.06
Academic burnout	14 (3)	6-point Likert	52.10 ± 8.90	26∼77	3.72 ± 0.64	1.86∼5.50	−0.08	−0.01

**Table 3 nursrep-15-00339-t003:** Model fit.

Model	Normed χ^2^	*p*	GFI	SRMR	RMSEA	TLI	NFI	CFI	AGFI
Criteria	≤3	<0.001	≥0.90	≤0.05	≤0.05	≥0.90	≥0.90	≥0.90	≥0.90
Hypothetical Model	51.80	<0.001	0.95	0.050	0.359	0.197	0.919	0.920	0.297
Modified model	1.71	0.182	0.997	0.020	0.042	0.989	0.995	0.998	0.974

**Table 4 nursrep-15-00339-t004:** Revised model. Effects of predictor variables in the path model (*N* = 396).

Endogenous Variables	Exogenous Variables	Standardized Direct Effect	Standardized Indirect Effect	Standardized Total Effect	SMC ^1^ (%)
		β (*p*)	β (*p*)	β (*p*)	
Academic engagement	Job-seeking stress	−0.116 (0.023)	−0.252 (<0.001)	−0.369 (<0.001)	0.372
Academic engagement	Academic self-efficacy	0.547 (<0.001)	-	0.547 (<0.001)	
Academic engagement	Professor trust	-	0.108 (<0.001)	0.108 (<0.001)	
Academic burnout	Job-seeking stress	0.315 (<0.001)	0.234 (<0.001)	0.549 (<0.001)	0.504
Academic burnout	Academic self-efficacy	−0.507 (<0.001)	-	−0.507 (<0.001)	
Academic burnout	Professor trust	-	−0.100 (<0.001)	−0.100 (<0.001)	
Academic self-efficacy	Job-seeking stress	−0.428 (<0.001)	−0.033 (0.001)	−0.461 (<0.001)	0.251
Academic self-efficacy	Professor trust	0.198 (<0.001)	0.108 (<0.001)	0.198 (< 0.001)	
Professor trust	Job-seeking stress	−0.169 (0.001)	-	−0.169 (0.001)	0.028

^1^ SMC = squared multiple correlation.

## Data Availability

The datasets used and/or analyzed during the current study available from the corresponding author on reasonable request.

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
