# Peer review of "Factors Influencing Nursing Students’ Academic Engagement and Burnout During the COVID-19 Pandemic: A Path Analysis"

_nursrep, 2025, doi:10.3390/nursrep15090339_

Round 1

Reviewer 1 Report

Comments and Suggestions for Authors

LInes 27, 28- Intro sentence (line 27 and 28) uses the word stress a lot, consider doing something like this, "...various sources of stress; academic, interpersonal, extracurricular.. "

Line 30-Data from KNCSA hospital- is that hiring per year? (line 30)

Line 32- could be worded clearer, it seems you are trying to say these are health related issues stemming from job-seeking stress but that isn't clear

Line 34 would actually be a better start of the paragraph- introducing the issues of nursing students and THEN talking about other types of stress.. just a thought.

Line 41- mentions the doctors strike increasing stress but is that because they are stressed out about the strike or because the strike has resulted in less open jobs?  please clarify

Line 43- talks about academic burnout- is this related to the job seeking stress?  I would try to clarify that there is a relationship between things that affect the nursing student in the first couple of sentences of the intro rather than just waiting until the last paragraph of the introduction.... seems like random sentences about a topic without connection.

Line 46, doesn't need a ';' or the word moreover

Line 81- hypothesis could be grouped a little more clearly, like all the job-seeking stress first (there are 4) then the trust in professors (there are 3) then the academic self-efficacy (there are 2).  The way it is presented is a little confusing.

Line 168- did they just do one questionnaire, or all of the different scales?  It seems that there were several tools used, did they all get put together?

Table 1 has a lot of information in it. If it is reported in the paragraph, no need to double represent in the table. Try to pull out anything that doesn't need to be in the table. I would pull out any of the characteristics that didn't have significant data and write a sentence about little or no correlation. OR, just write a paragraph about the most significant findings and correlations and let the table speak for itself.

Why is the age demographic <22, 22-23, and >23? I would write a sentence about why that is represented like that.

It is not clear from your measurements section that you were planning to compare the burn out and engagement with the demographics.  This is confusing. It is clear that you have a lot of data here but it may be better represented if you only presented in data format the demographics and then pulled out the significant correlations that you have made.  For example, the participants are divided by grade and the academic engagement and burnout are evaluated based on grade, but the grade level is not compared to the exercise level. A description or detail of why you chose to try to correlate some and not all would be helpful.

For table 2 it would be helpful to have the number of items listed per variable as well- for the job seeking stress there were 4 categories each with a 5 point scale, this could be made clearer on the table rather than just stating the mean score and the mean score for each item.

Line 219- this sentence isn't clear what you mean.. those who answered no to perceived stress? Also no addressing the degree of perceived interpersonal relationships that was mentioned at the start of the paragraph

Line 221- does this relate to the academic engagement?  Again, not clear

Line 227/228 females under 23.. it seems both the sex and age are grouped together rather than details about each characteristic.  If that is how you want to present it, then I would clarify that the rest of the groupings (male, other ages) weren't statistically significant

Line 335/336 are you saying that the restriction of social activities decreased the number of jobs? Would reword this.

Line 323- which studies showed that burnout was a predictor?

Line 327- putting a conclusion in the discussion section move this to the conclusion.

Line 349/350- this should be a suggestion, not a conclusion.  The study shows correlation but not causation so some of the actions that are being proposed are good for future growth but they are worded like 'if you do this, you will get this'.. (increase self efficacy, then increase engagement).  This study doesn't directly show that causation, just correlation.

Conclusion paragraph is confusing.  would reword.

Just because you have all the data doesn't mean it all has to be reported in the same article.  This article is very important and has interesting findings but would benefit from focusing on key details, or even just reporting on one of the factors in depth as opposed to trying to get all the data in for all of the tools used.  For example, use the demographics and just talk about the connection to burnout.  Or compare the self efficacy scores with that of the job- seeking stress.  It would even make sense to just pick a couple of the hypotheisis to focus on as opposed to all 9.  If you want to do all of them there should be more flushing out of each area.

Author Response

Response to Reviewer 1 Comments

Thank you very much for taking the time to review our manuscript. Please find the detailed responses below and the corresponding revisions/corrections highlighted in the re-submitted file.

1. Questions for General Evaluation

Reviewer’s Evaluation

Response and Revisions

Does the introduction provide sufficient background and include all relevant references?

Yes

Thank you very much for the positive feedback.

Is the research design appropriate?

Yes

Thank you very much for the positive feedback.

Are the methods adequately described?

Must be improved

We revised the methods section.

Are the results clearly presented?

Can be improved

We revised the results section.

Are the conclusions supported by the results?

Yes

Thank you very much for the positive feedback.

Are all figures and tables clear and well-presented?

Can be improved

We revised the tables. 

2. Point-by-point response to Comments and Suggestions for Authors

Comments 1: LInes 27, 28- Intro sentence (line 27 and 28) uses the word stress a lot, consider doing something like this, "...various sources of stress; academic, interpersonal, extracurricular.. "

Response 1: Thank you for pointing this out. We revised the sentence as your comment: Lines (26-27). “College students in South Korea have various sources of stress; academic, interpersonal, extracurricular, family-related, and job-related, but it is known that stress related to job preparation is the most stressful.”

Comments 2: Line 30-Data from KNCSA hospital- is that hiring per year? (line 30)

Response 2: Thank you for pointing this out. We revised the sentence as following for greater clarity. This revision can be found in line 44~48 of the revised manuscript. “In the first half of 2024, university hospitals faced severe financial difficulties due to the medical resident strike, resulting in almost no recruitment of new registered nurses. Despite an estimated 21,000 fourth-year nursing students nationwide, only one university hospital among the tertiary hospitals in the Seoul metropolitan area recruited new registered nurses during this period.”

Comments 3: Line 32- could be worded clearer, it seems you are trying to say these are health related issues stemming from job-seeking stress but that isn't clear.

Response 3: Thank you for pointing this out. We revised the sentence and this revision can be found in line 30~33 of the revised manuscript. “Job-seeking stress has been found to be associated with a variety of health-related problems, including tinnitus, nervousness, depression, gastrointestinal disturbances, insomnia, and social phobia, with approximately 70% of adults reportedly experiencing one or more of these symptoms.”

Comments 4: Line 34 would actually be a better start of the paragraph- introducing the issues of nursing students and THEN talking about other types of stress.. just a thought.

Response 4: Thank you for pointing this out. After discussion regarding the start of the paragraph, we decided to revise the sentence as follows and keep it in its original position. This revision can be found in line 34~37 of the revised manuscript. “To secure positions at highly competitive institutions, such as tertiary hospitals, Korean nursing students must dedicate substantial time and effort to achieving excellence in both theoretical coursework and clinical practicums. In addition, they are expected to cultivate proficiency in professional English and actively engage in volunteer activities and international training programs to enhance their professional competitiveness in an increasingly demanding job market”.

Comments 5: Line 41- mentions the doctors strike increasing stress but is that because they are stressed out about the strike or because the strike has resulted in less open jobs?  please clarify.

Response 5: Thank you for pointing this out. To improve clarity, we revised the sentences and these revisions can be found in line 40~49 of the revised manuscript. “More, in response to the South Korean government’s proposal to increase medical school admissions to address a doctor shortage, over 12,000 junior doctors walked out, medical professors joined in solidarity, and more than 95% of 2025 medical graduates are reportedly refusing to take the licensing exam [7]. In the first half of 2024, university hospitals faced severe financial difficulties due to the medical resident strike, re-sulting in almost no recruitment of new registered nurses. Despite an estimated 21,000 fourth-year nursing students nationwide, only one university hospital among the tertiary hospitals in the Seoul metropolitan area recruited new registered nurses during this period [3]. The doctors' strike has exacerbated job-seeking anxiety and stress among nursing students.”

Comments 6: Line 43- talks about academic burnout- is this related to the job seeking stress?  I would try to clarify that there is a relationship between things that affect the nursing student in the first couple of sentences of the intro rather than just waiting until the last paragraph of the introduction.... seems like random sentences about a topic without connection.

Response 6: Thank you for pointing this out. We incorporated this into the introduction and described it as follows in line 33~34. “This also affects students’ academic engagement and contributes to academic burnout.”

Comments 7: Line 46, doesn't need a ';' or the word moreover.

Response 7: Thank you for pointing this out. We removed both a ';' and the word moreover.  

Comments 8: Line 81- hypothesis could be grouped a little more clearly, like all the job-seeking stress first (there are 4) then the trust in professors (there are 3) then the academic self-efficacy (there are 2).  The way it is presented is a little confusing.

Response 8: Thank you for pointing this out. To enhance readability, the hypotheses were organized into groups in the following order: job-seeking stress (n = 4), academic self-efficacy (n = 2)., and faculty trust (n = 3). These revisions can be found in line 394~404 of the revised manuscript.

Group 1: Job-seeking stress

  1. Nursing students’ job-seeking stress will negatively affect academic engagement.
  2. Nursing students’ job-seeking stress will positively affect academic burnout.
  3. Nursing students’ job-seeking stress will negatively affect academic self-efficacy.
  4. Nursing students’ job-seeking stress will negatively affect professor trust.

Group 2: Academic self-efficacy

  1. Nursing students’ academic self-efficacy will positively affect academic engagement.
  2. Nursing students’ academic self-efficacy will negatively affect academic burn-out.

Group 3: Trust in professors

  1. Nursing students’ trust in professors will positively affect academic self-efficacy.
  2. Nursing students’ trust in professors will positively affect academic engagement.
  3. Nursing students’ trust in professors will negatively affect academic burnout.

Comments 9: Line 168- did they just do one questionnaire, or all of the different scales?  It seems that there were several tools used, did they all get put together?

Response 9: Thank you for pointing this out. We revised the sentence to enhance clarity. This revision can be found in line 177~179 of the revised manuscript. “After obtaining cooperation from faculty in each department, a survey pack-age—including the questionnaire, instructions for completion, and a return envelope—was either mailed or handed directly to participants.”

Comments 10: Table 1 has a lot of information in it. If it is reported in the paragraph, no need to double represent in the table. Try to pull out anything that doesn't need to be in the table. I would pull out any of the characteristics that didn't have significant data and write a sentence about little or no correlation. OR, just write a paragraph about the most significant findings and correlations and let the table speak for itself.

Response 10: Thank you for pointing this out. To ensure transparency and allow for potential meta-analysis, we presented all demographic variables in Table 1, but we revised a paragraph in line 196~202.

Comments 11: Why is the age demographic <22, 22-23, and >23? I would write a sentence about why that is represented like that.

Response 11: Thank you for pointing this out. The age groups (<22, 22–23, >23) were categorized based on the typical age distribution of nursing students across different academic years in South Korea, allowing for meaningful comparisons between underclassmen, upperclassmen, and older students who may have delayed entry or returned to school.

Comments 12: It is not clear from your measurements section that you were planning to compare the burn out and engagement with the demographics. This is confusing. It is clear that you have a lot of data here but it may be better represented if you only presented in data format the demographics and then pulled out the significant correlations that you have made. For example, the participants are divided by grade and the academic engagement and burnout are evaluated based on grade, but the grade level is not compared to the exercise level. A description or detail of why you chose to try to correlate some and not all would be helpful.

Response 12: Thank you for pointing this out. The competition for university admission among high school students in Korea is extremely intense. Therefore, students who are older but in lower grade levels are likely to be those with delayed entry or who have returned to school after a break. Considering the typical age range of Korean university students, we assumed that students who entered university immediately after high school without a break may have been more strongly affected by academic engagement and burnout due to the highly competitive admissions process. For this reason, we chose to control for this demographic variable and included it in the analysis. In addition, the aim of this study was not to explore correlations among all demographic variables but rather to evaluate the relationships between specific key variables. Therefore, we did not compare all possible combinations, such as grade level and exercise level. We hope this clarification addresses your concern.

Comments 13: For table 2 it would be helpful to have the number of items listed per variable as well- for the job seeking stress there were 4 categories each with a 5 point scale, this could be made clearer on the table rather than just stating the mean score and the mean score for each item.

Response : Thank you for pointing this out. In the revised manuscript, we have updated Table 2 to include the number of items for each variable, categories, and scale. We also added the explanation of the academic engagement instrument.

Comments 14: Line 219- this sentence isn't clear what you mean.. those who answered no to perceived stress? Also no addressing the degree of perceived interpersonal relationships that was mentioned at the start of the paragraph.

 Response 14: Thank you for pointing this out. We revised the sentence, and this revision can be found in line 220~223 of the revised manuscript. “However, post-hoc analysis revealed that differences in perceived stress levels were not statistically significant. Academic engagement was also significantly higher among participants who rated their interpersonal relationships as “very good” or “good”.  

Comments 15: Line 221- does this relate to the academic engagement? Again, not clear.

Response 15: Thank you for pointing this out. We revised sentence and this revision can be found in line 223~225 of the revised manuscript. “Regarding career plans after graduation, participants intending to work at a university hospital had significantly higher academic engagement compared to those planning to attend graduate school.”

Comments 16: Line 227/228 females under 23.. it seems both the sex and age are grouped together rather than details about each characteristic. If that is how you want to present it, then I would clarify that the rest of the groupings (male, other ages) weren't statistically significant.

Response 16: Thank you for pointing this out. We revised the sentence as following in line 230~231: “Academic burnout was significantly higher among female participants and those under 23 years of age compared to participants over 23.”

Comments 17: Line 335/336 are you saying that the restriction of social activities decreased the number of jobs? Would reword this.

Response 17: Thank you for pointing this out. We reworded the sentence, and this revision can be found in line 351~354 of the revised manuscript. “ Following the World Health Organization's declaration of COVID-19 as a global pandemic in March 2020, the labor market experienced a significant downturn, including a decline in nursing employment opportunities, which in turn intensified job-seeking stress among nursing students.”

Comments 18: Line 323- which studies showed that burnout was a predictor?

Response : Thank you for pointing this out. We revised the sentence and this revision can be found in line 339~342 of the revised manuscript. “A longitudinal prospective study conducted among Swedish nursing students found that academic burnout predicted lower learning engagement and job readiness during their studies, as well as reduced occupational skills and increased turnover intention one year after graduation.”

Comments 19: Line 327- putting a conclusion in the discussion section move this to the conclusion.

Response 19: Thank you for pointing this out. By revising the previous sentence, we decided to leave the following sentence as it is.

Comments 20: Line 349/350- this should be a suggestion, not a conclusion. The study shows correlation but not causation so some of the actions that are being proposed are good for future growth but they are worded like 'if you do this, you will get this'.. (increase self efficacy, then increase engagement). This study doesn't directly show that causation, just correlation.

Response 20: Thank you for pointing this out. We tried to mention our findings. To improve clarity in the writing, we revised the sentence as follows: “Our study revealed that academic self-efficacy among nursing students had a direct impact on both academic engagement and academic burnout.”

Comments 21: Conclusion paragraph is confusing. would reword.

Response 21: Thank you for pointing this out. We revised the conclusion section. “This study identified key factors influencing academic engagement and burnout among nursing students through path analysis. The proposed model demonstrated that job-seeking stress, academic self-efficacy, and trust in professors were significantly as-sociated with students’ levels of academic engagement and burnout. As academic burnout continues to increase, strategies to enhance engagement and reduce burnout are essential for supporting students’ successful adjustment to academic life and future clinical environments.

Based on these findings, it is necessary to develop and implement intervention programs that strengthen nursing students’ academic self-efficacy. Such programs could include elements like peer mentoring and stress management education. Enhancing academic self-efficacy may contribute to increased engagement and reduced burnout over time. Furthermore, the results highlight the importance of the professor-student relationship. It is recommended that nursing educators reflect on and refine their instructional strategies to build trust and foster a supportive learning environment. Ultimately, such efforts may enhance students’ academic self-efficacy, thereby increasing academic engagement and reducing burnout. Evaluating the effectiveness of these approaches will be essential in promoting sustainable academic performance and long-term professional development.”

Comments 22: Just because you have all the data doesn't mean it all has to be reported in the same article. This article is very important and has interesting findings but would benefit from focusing on key details, or even just reporting on one of the factors in depth as opposed to trying to get all the data in for all of the tools used. For example, use the demographics and just talk about the connection to burnout. Or compare the self efficacy scores with that of the job- seeking stress. It would even make sense to just pick a couple of the hypotheisis to focus on as opposed to all 9. If you want to do all of them there should be more flushing out of each area.

Response 22: Thank you for pointing this out. While we appreciate the editor’s feedback, we chose not to incorporate this particular suggestion.

Reviewer 2 Report

Comments and Suggestions for Authors

This study on nursing students’ academic engagement and burnout is well-structured and clearly presented.

Can the authors please simplify statistical interpretation: A brief explanation of key model fit indices (like RMSEA or CFI) in plain language could help readers less familiar with structural equation modelling.

line 252 - Can the authors explain why the overall model was good (or possibly revise this statement) since the normed overall fit (x-squared) was very high suggesting poor fit. The RMSEA is also a more acceptable demonstration of model fitness and appears to be very poor and problematic (presumably why correction was necessary).

Can the authors please expand on rejected hypotheses: A short discussion on why professor trust didn’t directly affect engagement or burnout would add helpful context.

Can the authors please enhance Methods Section by:
   - Discussing limitations of sampling and data collection.
   - Providing more context for the choice of statistical methods.
   - Clarifying procedures around missing data and participant recruitment.

Author Response

Response to Reviewer 2 Comments

Thank you very much for taking the time to review our manuscript. Please find the detailed responses below and the corresponding revisions/corrections highlighted in the re-submitted file.

1. Questions for General Evaluation

Reviewer’s Evaluation

Response and Revisions

Does the introduction provide sufficient background and include all relevant references?

Yes

Thank you very much for the positive feedback.

Is the research design appropriate?

Yes

Thank you very much for the positive feedback.

Are the methods adequately described?

Yes

Thank you very much for the positive feedback.

Are the results clearly presented?

Yes

Thank you very much for the positive feedback.

Are the conclusions supported by the results?

Yes

Thank you very much for the positive feedback.

Are all figures and tables clear and well-presented?

Yes

Thank you very much for the positive feedback

2. Point-by-point response to Comments and Suggestions for Authors

Comments 1: This study on nursing students’ academic engagement and burnout is well-structured and clearly presented.

Response 1: We deeply appreciate your positive feedback.

Comments 2: Can the authors please simplify statistical interpretation: A brief explanation of key model fit indices (like RMSEA or CFI) in plain language could help readers less familiar with structural equation modelling.

Response 2: We appreciate the reviewer’s suggestion. As recommended, we have included brief explanations of key model fit indices, such as RMSEA and CFI, in plain language to help readers who may not be familiar with structural equation modeling. These explanations have been added to enhance clarity and accessibility in line 250~266.

“To assess how well the proposed model fits the observed data, we examined a range of model fit indices. For absolute fit, we considered the Normed Chi-square (χ²/df), which evaluates model fit while accounting for model complexity—values be-low 3 are typically considered acceptable. We also included the Goodness of Fit Index (GFI), which indicates how closely the model reproduces the observed covariance matrix (values closer to 1 suggest better fit), and the Standardized Root Mean Square Residual (SRMR), which measures the average discrepancy between observed and predicted correlations—lower values indicate a better fit. Additionally, we used the Root Mean Square Error of Approximation (RMSEA) to assess how well the model would fit the population covariance matrix; values below 0.05 indicate close fit, and values up to 0.08 suggest reasonable fit. To evaluate incremental fit, we employed the Tucker-Lewis Index (TLI), Normed Fit Index (NFI), and Comparative Fit Index (CFI), which compare the target model with a baseline model, typically the null model. Values above 0.90 or 0.95 are generally considered to reflect a good fit. Lastly, for parsimonious fit, which considers both the goodness of fit and model simplicity, we examined the Adjusted Goodness of Fit Index (AGFI), a modified version of GFI that penalizes model complexity. Higher AGFI values indicate a better balance between fit and simplicity [25].”

Comments 3: line 252 - Can the authors explain why the overall model was good (or possibly revise this statement) since the normed overall fit (x-squared) was very high suggesting poor fit. The RMSEA is also a more acceptable demonstration of model fitness and appears to be very poor and problematic (presumably why correction was necessary).

Response 3: We appreciate the reviewer’s insightful comment. Upon revisiting our interpretation, we agree that the initial model’s overall fit was indeed problematic, as indicated by the high Normed χ² value (51.80) and particularly poor RMSEA (.359). Thus, we acknowledge that describing the initial model fit as "good" was inappropriate. We have revised this statement to clearly reflect the inadequacies of the initial model fit and emphasized that these poor indicators (high Normed χ² and RMSEA values) necessitated the modifications we subsequently performed. Furthermore, we have clarified in the manuscript that the modified model achieved acceptable fit indices (e.g., RMSEA = .042, Normed χ² = 1.71), thereby justifying the final model selection. These revisions can be found in line 267~281 of the revised manuscript. “Initially, the results indicated poor model fit, with Normed χ² = 51.80 (p < .001), GFI = .95, SRMR = .050, RMSEA = .359, TLI = .197, NFI = .919, CFI = .920, and AGFI = .297. The notably high Normed χ² and RMSEA values suggested substantial discrepancies between the hypothesized model and the data, requiring further adjustments. Given these poor fit indicators, we revised the model based on modification indices. Specifically, we deleted two insignificant paths: professor trust leading to academic burnout, and professor trust leading to academic participation. Additionally, correlations between the measurement errors of academic engagement and academic burnout, both endogenous latent variables, were established based on the modification indices.

After these adjustments, a difference test in model fit was conducted to compare the original hypothetical model and the modified model. The modified model showed a substantially improved and acceptable fit with Normed χ² = 1.71 (p = .182), GFI = .997, SRMR = .020, RMSEA = .042, TLI = .989, NFI = .995, CFI = .998, and AGFI = .974. These revised results indicated that the final modified model satisfactorily met recommended criteria for good fit (Table 3).”

Comments 4: Can the authors please expand on rejected hypotheses: A short discussion on why professor trust didn’t directly affect engagement or burnout would add helpful context.

Response 4: Thank you for pointing this out. We expanded on rejected hypotheses as following: “Hypotheses 8 and 9, which proposed that nursing students’ trust in professors would directly influence academic engagement and burnout, were not supported. A possible explanation is that professor trust does not directly affect engagement or burnout but instead operates indirectly through mediating factors such as academic self-efficacy. Indeed, our results showed that professor trust significantly predicted academic self-efficacy (β = .198, p < .001), which in turn strongly predicted engagement (β = .547, p < .001) and burnout (β = –.507, p < .001). These findings suggest that professor trust may be better understood as fostering students’ confidence in their academic abilities which subsequently shapes their engagement and susceptibility to burnout. This indirect relationship is consistent with prior research highlighting the mediating role of self-efficacy in educational contexts [32]. Accordingly, professor trust may be more accurately regarded as an antecedent to self-efficacy beliefs rather than a direct determinant of student outcomes. Further research is needed to examine additional psychological mediators or contextual moderators that may clarify these indirect effects.”

Comments 5: Can the authors please enhance Methods Section by:

- Discussing limitations of sampling and data collection.

- Providing more context for the choice of statistical methods.

- Clarifying procedures around missing data and participant recruitment.

Response 5: We thank the reviewer for the valuable suggestion. In the revised manuscript, we have enhanced the Methods section in several ways.

First, we added a discussion of potential limitations related to the sampling strategy and data collection procedures, acknowledging how these factors may influence the generalizability of our findings. These revisions can be found in line 394~404 of the revised manuscript. “This study has several methodological limitations that should be acknowledged. One limitation is that the use of convenience sampling from 11 nursing departments may limit the representativeness of the sample. Although participants were recruited from multiple institutions nationwide, the findings cannot be generalized to all nursing students in Korea. Another is because data collection relied on self-administered questionnaires distributed with the cooperation of faculty members, there may have been selection bias, as students who were more motivated or interested in the study topic may have been more likely to participate. Finally, the self-report nature of the survey may have introduced response bias or social desirability bias, which could affect the accuracy of the reported data. These limitations should be considered when interpreting the findings and assessing their generalizability.”   

Second, we provided further justification for the choice of statistical methods in data analysis and discussion, clarifying their appropriateness in addressing the research questions and data characteristics. These revisions can be found in line 187~194 and 406~411 of the revised manuscript.

“The collected data were analyzed using IBM SPSS Statistics Version 25 and AMOS Version 20. Descriptive statistics were used to summarize the participants’ general characteristics, as well as the distribution of key study variables. To examine differences in academic engagement and burnout across demographic subgroups, independent t-tests or one-way ANOVAs were applied, with Scheffé’s post-hoc test conducted when group differences were significant. These procedures were chosen to identify potential variations in outcomes according to categorical characteristics. Finally, path analysis was conducted to test the hypothesized model, as it allows simultaneous estimation of direct and indirect effects among multiple variables.”

“The use of path analysis was particularly appropriate for this study, as it enabled the simultaneous examination of direct and indirect effects among job-seeking stress, academic self-efficacy, professor trust, academic engagement, and academic burnout. This approach provided a more comprehensive understanding of the interrelationships among the study variables than would have been possible with simpler statistical techniques.”

Finally, we elaborated on procedures concerning missing data and participant recruitment to ensure greater transparency and reproducibility. These revisions can be found in line 101~106 and 176~183 of the revised manuscript.

“The participants were nursing students enrolled in 4-year nursing programs across 11 nursing departments nationwide. The minimum sample size required for correlation analysis with a significance level of .05, power of .90, and a medium effect size (0.20) was calculated to be 258 using G*Power 3.1.9.2. A total of 450 questionnaires were distributed, and 417 were returned (92.7%). After excluding four questionnaires with incomplete responses, 413 were retained for the final analysis, yielding a valid response rate of 91.8%.”

“Data were collected between August 30 and December 13, 2021, using convenience sampling in the 11 nursing departments. After obtaining cooperation from faculty in each department, a survey package—including the questionnaire, instructions for completion, and a return envelope—was either mailed or handed directly to participants. Students who consented to participate were asked to complete the survey, seal it in the envelope, and return it to the research team. Completion of the questionnaire required approximately 20 minutes. Returned surveys were screened for completeness, and only fully completed questionnaires were included in the analysis.”

Round 2

Reviewer 1 Report

Comments and Suggestions for Authors

Much better. Thank you for all of your hard work- this is a good article